# Luminescent Properties and Charge Compensator Effects of SrMo_0.5_W_0.5_O_4_:Eu^3+^ for White Light LEDs

**DOI:** 10.3390/molecules28062681

**Published:** 2023-03-16

**Authors:** Li Kong, Hao Sun, Yuhao Nie, Yue Yan, Runze Wang, Qin Ding, Shuang Zhang, Haihui Yu, Guoyan Luan

**Affiliations:** 1Institute of Petrochemical Technology, Jilin Institute of Chemical Technology, Jilin 132022, China; 2School of Chemical Engineering, Northeast Electric Power University, Jilin 132012, China

**Keywords:** SrMo_0.5_W_0.5_O_4_:Eu^3+^, charge compensator, Luminescence performance, w-LED

## Abstract

The high-temperature solid-phase approach was used to synthesize Eu^3+^-doped SrMo_0.5_W_0.5_O_4_ phosphors, whose morphological structure and luminescence properties were then characterized by XRD, SEM, FT-IR, excitation spectra, emission spectra, and fluorescence decay curves. The results reveal that the best phosphor synthesis temperature was 900 °C and that the doping of Eu^3+^ and charge compensators (K^+^, Li^+^, Na^+^, NH_4_^+^) had no effect on the crystal phase change. SrMo_0.5_W_0.5_O_4_:Eu^3+^ has major excitation peaks at 273 nm, 397 nm, and 464 nm, and a main emission peak at 615 nm, making it a potential red fluorescent material to be used as a down converter in UV LEDs (273 nm and 397 nm) and blue light LEDs (464 nm) to achieve Red emission. The emission spectra of Sr_1−y_Mo_0.5_W_0.5_O_4_:yEu^3+^(y = 0.005, 0.01, 0.02, 0.05, 0.07) excited at 273 were depicted, with the Eu^3+^ concentration increasing the luminescence intensity first increases and then decreases, the emission peak intensity of SrMo_0.5_W_0.5_O_4_:Eu^3+^ achieves its maximum when the doping concentration of Eu^3+^ is 1%, and the critical transfer distance is calculated as 25.57 Å. When various charge compensators such as K^+^, Li^+^, Na^+^, and NH_4_^+^ are added to SrMo_0.5_W_0.5_O_4_:Eu^3+^, the NH_4_^+^ shows the best effect with the optimal doping concentration of 3wt%. The SrMo_0.5_W_0.5_O_4_:Eu^3+^,NH_4_^+^ color coordinate is (0.656,0.343), which is close to that of the ideal red light (0.670,0.333).

## 1. Introduction

As a new generation of the light source of solid-state lighting, white light-emitting diodes (hereafter referred to as the white light LEDs, w-LEDs, etc.) have piqued the interest of scholars both at home and abroad for their high efficiency, energy savings, and environmental protection advantages [1,2,3,4]. The white light LED used to be created by combining a GaN chip that emits blue light with yellow phosphors (YAG:Ce^3+^) that can be effectively excited by blue light [5]. However, this approach typically generates a low color rendering index because of the lack of red light in the emission spectrum of the yellow phosphor. The solution is to add red phosphors that can be efficiently excited by blue light [6,7] or use the high-efficiency UV LED and the phosphors that can be excited by it [8,9]. As a result, it is critical to investigate red phosphors that can be successfully stimulated by blue light and UV light.

It is well known that Eu^3+^ is an outstanding rare earth ion generating red light and can be effectively stimulated by blue light and UV light [10,11,12,13,14]. For instance, a new red phosphor Sr_3_NaSbO_6_:Eu^3+^ doped with Eu^3+^ was developed, and its emission spectra under excitation at 285 nm is located 500–700 nm, with the primary peak at 618 nm, indicating that this phosphor is a red phosphor that can be successfully stimulated by UV light [15]. Li_2.06_Nb_0.18_Ti_0.76_O_3_:Eu^3+^ phosphors by sol-gel method were prepared. When the doping proportion of Eu^3+^ is x = 3 wt%, the primary excitation peak is at 396 nm, the central emission peak is at 612 nm, and its color coordinate is better than the commercial red phosphor Y_2_O_3_:Eu^3+^ [16]. The phosphors Y_2_SiO_5_:Eu^3+^ synthesized by the solid-state reaction method can be effectively excited by near-UV (394 nm), and the major peak is located at 611 nm, the critical quenching concentration of Eu^3+^ in the phosphor is determined to be 15 mol%, and the critical transfer distance is calculated as 8.90 Å; co-doping Y_2_SiO_5_:Eu^3+^ with Ge^4+^ helps to improve the luminescence intensity and color purity, it can be concluded that efficient red light emitting diodes were fabricated using Ge^4+^, Eu^3+^ co-doped phosphor based on near ultraviolet(NUV) excited LED lights [17]. By using a high-temperature solid-phase reaction, a new lithium salt type NaBaBi_2_(PO_4_)_3_:Eu^3+^ phosphor was synthesized, which can emit a main peak at 611 nm under the effective excitation of near UV and blue light, the color temperature and color purity are about 1800K and 88%, respectively, making it an excellent red, warm light material [18]. Eu^3+^-doped BaLaWO_7_ and SrLa_2_WO_7_ red phosphors were synthesized using the traditional solid-state reaction method [19].

Based on their low phonon energy, outstanding chemical and physical properties, good thermal stability, and strong charge transfer zone in the ultraviolet region, tungstates and molybdates have been widely employed as host materials to phosphors [20,21]. A highly uniform spindle-shaped SrMoO_4_:Eu^3+^ phosphor was developed, which produces the Eu^3+^ characteristic transition peak ^5^D_0_-^7^F_J_ (J = 1, 2, 3, 4) under ultraviolet light excitation (287 nm), with the ^5^D_0_-^7^F_2_ transition (613 nm) in the red region being the strongest [22]. The produced SrMoO_4_:Eu^3+^ phosphors synthesized by Yanan Zhu et al. can be successfully activated by ultraviolet light at 396 nm and emit red light with a prominent peak at 616 nm [23]. Dy^3+^-doped SrMoO_4_ nanophosphors were synthesized, which emit blue light at 485 nm and bright yellow light at 576 nm Under UV illumination at 353 nm [24]. SrWO_4_:Eu^3+^ phosphor was synthesized using the microwave radiation heating approach. The phosphor’s excitation spectrum falls in a strong absorption band centered at 295 nm and two weak sharp peaks centered at 389 and 467 nm, and the primary peaks of its emission spectra are positioned at 589 nm and 616 nm [25]. SrWO_4_:Eu^3+^ phosphors have been successfully synthesized, with the most substantial emission peaks in the emission spectrum at 615 nm under near UV (394 nm) and blue light (450 nm) excitation [26]. The emission intensity of CaW_0.4_MoO_4_:Eu^3+^ red phosphor is estimated to be 8.3 times that of CaWO_4_:Eu^3+^ phosphor [27]. It was discovered that adding Mo(VI) ions to the red phosphor Sr_2_ZnWO_6_:Eu^3+^ red phosphor significantly increased the emission intensity [28]. The phosphor Ca_0.3_Sr_0.7−1.5y−1.5z_Mo_1−x_W_x_O_4_:Eu_y_Lu_z_ was synthesized, and the most incredible emission intensity was observed at x = 0.2, y = 0.1 and z = 0.1 [29]. Gd_2(1−x)_Eu_2_x(Mo_y_W_1−_yO_4_)_3_ phosphors were synthesized, and its highest emission intensities increased with more W(VI) [30]. Despite a significant number of reports on tungstate-molybdate phosphors, there are fewer on SrMo_0.5_W_0.5_O_4_:Eu^3+^.

This research synthesized the red phosphors SrWO_4_:Eu^3+^, SrMo_0.5_W_0.5_O_4_:Eu^3+^, and SrMoO_4_:Eu^3+^ using a high-temperature solid-phase technique. Moreover, it examines their spectrum properties as well as the effect of different charge compensators on the luminescence properties of SrMo_0.5_W_0.5_O_4_:Eu^3+^.

## 2. Results and Discussion

### 2.1. Physical and Chemical Phase Analysis

Figure 1 reveals the X-ray powder diffraction (XRD) patterns of (a) SrMoO_4_, (b) SrMo_0.5_W_0.5_O_4_, and (c) SrWO_4_ synthesized at different temperatures. Figure 1a shows that the XRD patterns’ peak positions and relative intensities of the XRD patterns of the sample SrMoO_4_ at temperatures of 850 °C, 900 °C, 950 °C, and 1000 °C are essentially the same, which is consistent with the standard card of SrMoO_4_ (JCPDS 08-0482), indicating that the synthesized samples have a tetragonal crystal system with space group I41/a, and its unit cell data are a = b = 5.3909 Å, c = 12.0118 Å and α = β = γ = 90°. Strontium molybdate can be synthesized at these temperatures without forming an impurity phase. Furthermore, the highest peak intensity was discovered in the sample synthesized at 900 °C, indicating that the crystallinity of the sample is better at this temperature. As a result, the temperature to synthesize SrMoO_4_ is set to 900 °C. Figure 1b displays that the XRD patterns’ peak positions and relative intensities of the XRD patterns of the sample SrMo_0.5_W_0.5_O_4_ at those temperatures are essentially consistent with the standard card of SrMoO_4_ (JCPDS 08-0482), demonstrating that the synthesized samples have the structure of SrMoO_4_, and no new phase is formed. Due to the lanthanide contraction, the atomic and ionic radii of Mo and W, the second and third transition elements in the same group are very close (the atomic radii of Mo and W are both 139 pm, and the ionic radii of Mo(VI) and W(VI) are 59 pm and 60 pm, respectively), and their properties are quite similar. Besides, the structures of MoO_4_^2−^ and WO_4_^2−^ are the same. As a result, WO_4_^2−^ can easily replace MoO_4_^2−^ to form a solid solution. The peak intensity of the XRD pattern of SrMo_0.5_W_0.5_O_4_ at 900 °C is higher, indicating that the sample’s crystallinity is better at this temperature. As a result, 900 °C is the optimal synthesis temperature for SrMo_0.5_W_0.5_O_4_. Figure 1c shows that the XRD patterns of SrWO_4_ synthesized at temperatures of 850 °C, 900 °C, 950 °C, and 1000 °C are consistent with the standard card of SrWO_4_ (JCPDS 08-0490), indicating that the synthesized samples have a tetragonal crystal structure with the space group is I41/a (88), and that can synthesize pure phase strontium tungstate at these temperatures. Because the XRD peak of SrWO_4_ synthesized at 900 °C is the strongest, 900 °C is the best SrWO_4_ synthesis temperature.

Figure 2a shows that the diffraction peaks of the Sr_1−x_Mo_0.5_W_0.5_O_4_:xEu^3+^ XRD pattern are in line with the standard card #JCPDS 08-0482 (SrMoO_4_), indicating that the doping of Eu^3+^ in the SrMo_0.5_W_0.5_O_4_ system did not cause phase change and no new phase was created. Rare earth metal Eu and alkaline-earth metal Sr have similar atomic and ionic radii (the atomic radii of Eu and Sr are 208 pm and 215 pm, respectively, while the ionic radii of Eu^3+^ and Sr^2+^ are 112 pm and 94.7 pm, respectively). When Eu^3+^ is doped into the SrMo_0.5_W_0.5_O_4_ system, it takes the position of Sr^2+^ and creates a continuous solid solution. It has been reported that the O^2−^ is created in the system due to the imbalance in electrovalence as a result of the unequal substitution of Sr^2+^ with Eu^3+^ [23]. According to Figure 2b, the diffraction peaks of the XRD pattern of the phosphor Sr_0.99_MoO_4_:0.01Eu^3+^ are compatible with the standard card #JCPDS 08-0482 (SrMoO_4_), indicating that the sample forms pure phase SrMoO_4_, and no additional phases are created. That is to say, 1% Eu^3+^ can be added to SrMoO_4_ without generating a phase shift. As seen in Figure 2c, the diffraction peaks of the XRD pattern of the phosphor Sr_0.99_WO_4_:0.01Eu^3+^ are consistent with the standard card #JCPDS 08-0490 (SrWO_4_), indicating that the pure phase can still be obtained formed by doping 1% Eu^3+^ in SrWO_4_, and no additional substances form.

The electrovalent imbalance induced by the unequal substitution of Sr^2+^ with Eu^3+^ in the Sr_0.99_Mo_0.5_W_0.5_O_4_:0.01Eu^3+^ system can be rectified by adding charge compensators [31]. Figure 3 depicts the XRD patterns of SrMo_0.5_W_0.5_O_4_:Eu^3+^ after doping with various charge compensators. Figure 3 shows that the XRD diffraction peaks of SrMo_0.5_W_0.5_O_4_: Eu^3+^ after adding charge compensator K_2_CO_3_, Li_2_CO_3_, Na_2_CO_3_, NH_4_Cl are essentially consistent with the standard card of SrMoO_4_ (JCPDS 08-0482), that is, there is no charge in the lattice of SrMo_0.5_W_0.5_O_4_:Eu^3+^, and the phase is still SrMo_0.5_W_0.5_O_4_.

SrMo_0.5_W_0.5_O_4_ has a tetragonal crystal system with a scheelite structure, and each of its units contains one Sr site, one Mo/W site, and four O sites. According to Figure 4, there is only one type of cationic site, Sr, in the lattice, and each, on average, has eight coordinated oxygen ions which include four MoO_4_^2−^/WO_4_^2−^ that belong to the S4 symmetry and have no inversion center. Each central W/Mo site is coordinated with four identical O, forming a MoO_4_^2−^/WO_4_^2−^ tetrahedron. As the MoO_4_^2−^/WO_4_^2−^ tetrahedral configuration is quite stable, SrMo_0.5_W_0.5_O_4_ retains its lattice structure when Sr^2+^ is replaced by Eu^3+^.

The FT-IR spectrum of the sample SrMo_0.5_W_0.5_O_4_ was obtained by the KBr pressed disc method. As shown in Figure 5, the FT-IR spectra of the prepared samples have absorption peaks at 818 cm^−1^, 1630 cm^−1^, and 3420 cm^−1^, where the absorption peak at 818 cm^−1^ corresponds to the stretching vibration of O-W/Mo-O, indicating the existence of WO_4_^2−^ and MoO_4_^2−^ groups in the prepared samples. The absorption peaks at 1630 cm^−1^ and 3420 cm^−1^ are respectively attributed to the bending and stretching vibrations of O-H, causing the water vapor on the surface of the SrMo_0.5_W_0.5_O_4_ surface sample.

Figure 6 shows the SEM photos of the phosphor Sr_0.99_Mo_0.5_W_0.5_O_4_:Eu^3+^ synthesized using a high-temperature solid phase technique at 900 °C. The phosphor Sr_0.99_Mo_0.5_W_0.5_O_4_:0.01Eu^3+^ has sharp edges and corners, an irregular form, and a particle size of around 2 μm, with agglomeration produced by high-temperature solid-phase preparation.

### 2.2. Analysis of Luminescence Performance

Figure 7a–f show the excitation spectra of SrWO_4_:Eu^3+^, SrMo_0.5_W_0.5_O_4_: Eu^3+^, and SrMoO_4_:Eu^3+^ at 615 nm, and the emission spectra at 273 nm, respectively. Figure 7a,c,e show that the phosphors SrWO_4_:Eu^3+^, SrMo_0.5_W_0.5_O_4_: Eu^3+^, and SrMoO_4_:Eu^3+^ have a solid and broad CT band in the range of 200 nm to 330 nm, with the center wavelength of 273 nm. Furthermore, the f-f characteristic absorption peaks of Eu^3+^were also observed at 362 nm (^7^F_0_→^5^D_4_), 378 nm (^7^F_0_→^5^G_2_), 383 nm (^7^F_0_→^5^G_3_), 394 nm (^7^F_0_→^5^L_6_), 416 nm (^7^F_0_→^5^D_3_), 464 nm (^7^F_0_→^5^D_2_) and 534 nm (^7^F_0_→^5^D_1_); the peaks at 273 nm, 394 nm, and 464 nm are stronger, indicating that SrWO_4_:Eu^3+^, SrMo_0.5_W_0.5_O_4_: Eu^3+^, Figure 7b,d,f demonstrate that the emission spectra of SrWO_4_:Eu^3+^, SrMo_0.5_W_0.5_O_4_:Eu^3+^, and SrMoO_4_:Eu^3+^ are composed of a succession of sharp fronts, with several emission peaks detected at 568 nm, 591 nm, 615 nm, and 653 nm, corresponding to the ^5^D_0_→^7^F_0_, ^5^D_0_→^7^F_1_, ^5^D_0_→^7^F_2_, ^5^D_0_→^7^F_3_ transitions of Eu^3+^, respectively. When Eu^3+^ ions occupy the matrix’s inversion symmetry center site, the magnetic dipole transition of ^5^D_0_→^7^F_1_ prevails; conversely, Eu^3+^ ions occupy the matrix’s non-inversion symmetry center site, Eu^3+^ electric dipole transition of ^5^D_0_→^7^F_2_ dominates. In addition, the red emission peak corresponding to the ^5^D_0_→^7^F_2_ transition is the strongest, implying that Eu^3+^ is located in the non-inversion symmetry center lattice site of host lattices of SrWO_4_:Eu^3+^, SrMo_0.5_W_0.5_O_4_: Eu^3+^, and SrMoO_4_:Eu^3+^. So SrMoO_4_:Eu^3+^ can be used as a down converter in UV LEDs and blue light LEDs to achieve red emission.

Figure 8 depicts the emission spectra of SrWO_4_:0.01Eu^3+^, SrMo_0.5_W_0.5_O_4_:0.01Eu^3+^, and SrMoO_4_:0.01Eu^3+^ under 273 nm monitoring. The emission peak shape and position of Eu^3+^ ions remain constant across all samples. The intensity of the emission increases after the addition of Mo(VI) ions to SrWO_4_:Eu^3+^ and decreases as Mo(VI) ions totally replace the W(VI), and the emission spectrum of Sr_0.99_Mo_0.5_W_0.5_O_4_:0.01Eu^3+^ is the strongest. The reason for that is: the introduction of Mo(VI) ions will form MoO_4_^2−^ groups, which can efficiently modulate the diversity of the Eu^3+^ surrounding environment and shift the symmetry of the local crystal field, thereby promoting the charge transfer transition of O^2−^→Eu^3+^, the Eu^3+^ hypersensitive transition, and the electron-migration energy of MoO_4_^2−^ (M = W, Mo) in the matrix to transfer to Eu^3+^ [32]. Furthermore, after introducing Mo(VI), the average distance between WO_4_ groups becomes wider [27], leading to a lower energy transfer between WO_4_ groups and then more incident energy will be transferred to Eu^3+^. When the Mo(VI) ion concentration is too high, the impact of the ion-pair interaction between Eu^3+^ ions will be increased, leading to a reduction in the phosphor’s luminous efficiency [33]. Therefore, inserting Mo(VI) can effectively improve the luminous properties of SrWO_4_:Eu^3+^ phosphors.

Figure 9a depicts the emission spectra of SrMo_0.5_W_0.5_O_4_:Eu^3+^ with varying Eu^3+^ concentrations excited at 273 nm. Figure 9a shows that all samples’ peak forms and positions remain constant. However, with the Eu^3+^ concentration increasing, the luminescence intensity first increases and then decreases. The emission peak intensity of Sr_0.99_Mo_0.5_W_0.5_O_4_:0.01Eu^3+^ achieves its maximum when the doping concentration of Eu^3+^ is 1%, and if the concentration of Eu^3+^ continues to increase, the phenomenon of concentration quenching appears. This is because although the transition of emitted light increases with the increase of the Eu^3+^ concentration, which can effectively improve the intensity of the emitted light, the continuous increase of the doping amount of Eu^3+^ will narrow the distance between Eu^3+^, resulting in a decrease in emission intensity due to nonradiative energy transfer between Eu^3+^. To look into the energy transfer of Eu^3+^ ions in SrMo_0.5_W_0.5_O_4_, the critical distance of Eu^3+^ ions is first estimated using the formula below.

The critical distance *R_c_* can be computed using the Blass theory formula [34]:$$R_{c}=2\left(\frac{3V}{4\pi x_{cN}}\right)$$

In this equation, *V* denotes the unit cell volume, *X_c_* is the critical concentration of Eu^3+^ in SrMo_0.5_W_0.5_O_4_(the optimal doping concentration), and *N* denotes the number of cations per unit cell of SrMo_0.5_W_0.5_O_4_ crystal. Figure 9a shows the critical threshold concentration of Eu^3+^ is 0.01 in SrMo_0.5_W_0.5_O_4_ crystal, N = 4, V = 349.78 Å^3^. According to the Blass formula, *R_c_* = 25.57 Å. In general, non-radiative energy transfer modes are broadly classified as electron exchange interaction and electric multipole interaction. When the critical distance *R_c_* is around 5 Å, the non-radiative energy transfer mode is electron exchange interaction. When *R_c_* reaches 25.57 Å, much more than 5 Å, the energy transfer between Eu^3+^ in SrMo_0.5_W_0.5_O_4_: Eu^3+^ is electric multipolar interaction.

The energy transfer formula for the electric multipole interaction can be derived using Van Uitert’s theory [35]:$$\frac{I}{X}=K{\left[1+\beta {\left(X\right)}^{\frac{\theta }{3}}\right]}^{-1}$$

In this formula, *I* is the integrated emission intensity, *X* is the activator concentration above the critical concentration, and *K* and *β* are constants for a given matrix. Analyzing the constant θ confirms the energy transfer mode of the electric multipole interaction, and the number of cations in the unit cell of SrMo_0.5_W_0.5_O_4_ crystal can be deduced. θ = 6, 8, and 10 correspond to dipole-dipole (d-d), dipole-quaternary (d-q), and quaternary-quaternary (q-q) interactions, respectively. Figure 9b reveals the connection between log(I/X) and log(X) of SrMo_0.5_W_0.5_O_4_: Eu^3+^. If the slope –1.64 is –θ/3, then θ will be 4.92, which the value is closer to 6. As a result, the electric dipole-electric dipole (d-d) interaction causes the quenching concentration in Sr_1−x_Mo_0.5_W_0.5_O_4_:xEu^3+^.

The partial substitution of Sr^2+^ by Eu^3+^ in SrMo_0.5_W_0.5_O_4_:Eu^3+^ will result in a charge imbalance, leading to excessive charge defects in the lattice and thus decreasing the phosphor luminous efficiency. However, adding the right amount of good charge compensator can increase the sample’s luminous efficiency [31]. Figure 10 depicts the emission spectra of phosphors SrMo_0.5_W_0.5_O_4_: Eu^3+^, M (M = K^+^, Li^+^, Na^+^, NH_4_^+^) doped with various charge compensators. The addition of the charge compensator doesn’t modify the position of the emission peak of SrMo_0.5_W_0.5_O_4_: Eu^3+^. Various charge compensators have different effects on the luminescence intensity of SrMo_0.5_W_0.5_O_4_:Eu^3+^, but their doping will improve the luminescence intensity, with NH_4_^+^ having the best effect.

Figure 11 depicts the luminescence intensity of Sr_0.99_Mo_0.5_W_0.5_O_4_:0.01Eu^3+^ at various NH_4_^+^ doping concentrations (0%, 3%, 6%, 10%, 15%). The figure shows that when the concentration of NH_4_^+^ is low, the luminescence intensity of the sample increases as the concentration of NH_4_^+^ increases. The sample’s emission peak intensity reaches its maximum highest when the NH_4_^+^ doping concentration is 3%. As the concentration of NH_4_^+^ continues to increase, concentration quenching will occur. This is due to the fact that when the concentration of NH_4_^+^ is low, NH_4_^+^ can replace the position of Sr^2+^ in the lattice, lowering the symmetry of the lattice and modifying the local crystal field environment around Eu^3+^, which eventually increases the sample’s luminescence performance [36,37]; At the same time, due to the difference in the quantities of electric charges of NH_4_^+^ and Sr^3+^, oxygen vacancies will be formed after replacing Sr^2+^ in order to maintain the electrical neutrality of NH_4_^+^. These oxygen vacancies can transfer charge with Eu^3+^ [34], thereby increasing the sample’s luminescence intensity. On the other hand, the excess NH_4_^+^ will enter the lattice gaps and induce lattice distortions, affecting the luminescence intensity of the samples.

Figure 12 shows the luminescence decay curves of SrMo_0.5_W_0.5_O_4_:Eu^3+^ phosphors doped with several charge compensators (K^+^, Li^+^, Na^+^, NH_4_^+^) at an excitation wavelength of 464 nm and an emission wavelength of 615 nm. As illustrated in Figure 12, the decay curves of all samples’ emitted light satisfy a bi-exponential equation [38]:$$I\left(t\right)=I_{0}+A_{1}\exp\left(\frac{-t}{\tau_{1}}\right)+A_{2}\exp\left(\frac{-t}{\tau_{2}}\right)$$

In the formula, I(*t*) denotes the emission intensity at time t, I_0_ represents the initial emission intensity, A_1_ and A_2_ are the pre-exponential factors of each decay component, and τ_1_ and τ_2_ are the decay times of each component. The average emission decay time (τ_ave_) can be calculated using the below [38].
$$\tau_{\mathrm{ave}}=\frac{A_{1}\tau_{1}^{2}+A_{2}\tau_{2}^{2}}{A_{1}\tau_{1}+A_{2}\tau_{2}}$$

The average emission decay time τ_ave_ shown in Figure 12, was calculated to be 0.57 ms for Sr_0.99_Mo_0.5_W_0.5_O_4_:0.01Eu^3+^ and 0.0.51, 0.0.57, 0.56, and 0.0.58 ms for Sr_0.99_Mo_0.5_W_0.5_O_4_:0.01Eu^3+^, A (A = Li^+^, Na^+^, K^+^, NH_4_^+^), respectively. The emission decay times of all ceramic samples were very similar and slightly lower than that of the powder sample. This suggests that, in the ceramic samples, the electronic relaxation time from the split ^5^D_2_ energy levels to the lowest transition energy level ^5^D_0_ was reduced. When the charge compensator NH_4_^+^ concentration is 3% in the Sr_0.99_Mo_0.5_W_0.5_O_4_:0.01Eu^3+^, A (A = K^+^, Li^+^, Na^+^, NH_4_^+^) system, the fluorescence lifespan of the sample achieves a maximum of 0.58ms. It is also demonstrated that adding NH_4_^+^ can significantly improve the luminescent characteristics of the samples.

Figure 13 depicts the color coordinates of samples Sr_0.99_Mo_0.5_W_0.5_O_4_: 0.01Eu^3+^ (b), Sr_0.99_Mo_0.5_W_0.5_O_4_: 0.01Eu^3+^, 0.03NH_4_^+^ (c), where the color coordinates of b (0.642, 0.358) and c (0.656, 0.343) are both positioned at the edge of the red area, indicating that the synthetic samples have a high color purity. The color coordinates of the SrMo_0.5_W_0.5_O_4_: Eu^3+^ sample show a red-shifted after adding the charge compensator NH_4_^+^, demonstrating that NH_4_^+^ can successfully improve the luminescence properties of the non-SrMo_0.5_W_0.5_O_4_: Eu^3+^ sample. The coordinates are close to the ideal red light’s coordinates (0.670, 0.333) (d) and better than the commercial red phosphor Y_2_O_2_S:Eu^2+^’s coordinates (0.622, 0.351) (a).

## 3. Materials and Methods

### 3.1. Sample Preparation

All samples were synthesized in an air atmosphere using a high-temperature solid phase method. The raw materials included SrCO_3_(A.R.), MoO_3_(A.R.), WO_3_(A.R.), Eu_2_O_3_ (99.99%), Na_2_CO_3_ (A.R.), Li_2_CO_3_(A.R.), K_2_CO_3_ (A.R.), and NH_4_Cl (A.R.). They were accurately weighed based on the stoichiometric ratio of Sr_(1−y)_Mo_x_W_1−x_O_4_:yEu^3+^, transferred to an agate mortar, added a tiny amount of anhydrous ethanol, ground for 30 min, then transferred the blended powder was to a high-temperature furnace and calcined at a certain temperature for 5 h.

### 3.2. Sample Testing and Characterization

The structures of the samples were studied using a Bruker (Billerica, MA, USA) AXS D8 X-ray diffractometer (XRD), with Cu Kα lines as a radiation source. An operating voltage of 40 KV, An operating current of 30 mA, and a scanning range of 2θ = 15–80°; the microscopic morphology of the samples was characterized using a JSM-6490LV scanning electron microscope (SEM). The sample’s Fourier transform infrared (FT-IR) spectra were evaluated by a Perkin Elmer(Norwalk, CT, USA) Type NicoLet670-shaped Fourier transform infrared spectrometer using the KBr pressed-disc technique and a resolution of 4 cm^−1^. The excitation, and emission spectra, luminescence decay curves of the luminous material were evaluated using an Edinburgh FS5 fluorescence spectrometer equipped with a 150 W xenon lamp as an excitation light source. All of the preceding experiments were carried out at room temperature.

## 4. Conclusions

The phosphor SrMo_0.5_W_0.5_O_4_: Eu^3+^ synthesized by the high-temperature solid-phase technique has an optimal synthesis temperature of 900 °C. The phosphor possesses a tetragonal crystal structure, and the doping of Eu^3+^ does not affect the crystal phase. The phosphor’s Fourier infrared spectrum indicates a stretching vibration of O-W/Mo-O; the SEM shows irregular particles with sharp edges and corners, a particle size of 2 μm, and numerous agglomerations. The primary excitation peaks of the phosphor SrMo_0.5_W_0.5_O_4_: Eu^3+^ are positioned at 273 nm, 397 nm, and 464 nm, respectively, and are attributable to the charge migration of O^2−^→Eu^3+^ and the distinctive spectrum of Eu^3+^ (^7^F_0_→^5^L_6_, ^7^F_0_→^5^D_2_). The central light peak of the emitted light is around 615 nm, representing a possible red fluorescent material that can be used as a down converter in UV LEDs and blue light LEDs. When the emission spectra of red phosphors SrWO_4_:Eu^3+^, SrMo_0.5_W_0.5_O_4_: Eu^3+^, and SrMoO_4_:Eu^3+^ are compared, it is discovered that after the introduction of Mo(VI) into SrWO_4_:Eu^3+^, the emission intensity increases and when Mo(VI) ions completely replace the W(VI), the emission intensity decreases. The optimal doping concentration (quenching concentration)of Eu^3+^ in SrMo_0.5_W_0.5_O_4_: Eu^3+^ is 1%, and the quenching concentration is 1%, which is due to the galvanic-even-order interaction in the electric multilevel interaction, and its critical distance is R_c_ = 25.57 Å. Charge compensators of various types, including Eu^3+^: K^+^, Li^+^, Na^+^, and NH_4_^+^, were added into SrMo_0.5_W_0.5_O_4,_ with NH_4_^+^ having the best effect and the best doping concentration of it being 3%. In comparison to the color coordinates (0.642, 0.358) of SrMo_0.5_W_0.5_O_4_: Eu^3+^, the color coordinates (0.656, 0.343) of SrMo_0.5_W_0.5_O_4_: Eu^3+^, NH_4_^+^ exhibit an apparent red shift phenomenon, and the color coordinates of the obtained phosphors are all better than the color coordinates of commercial red phosphors (0.622, 0.351) and are closer to the ideal.

## Figures and Tables

**Figure 1 molecules-28-02681-f001:**
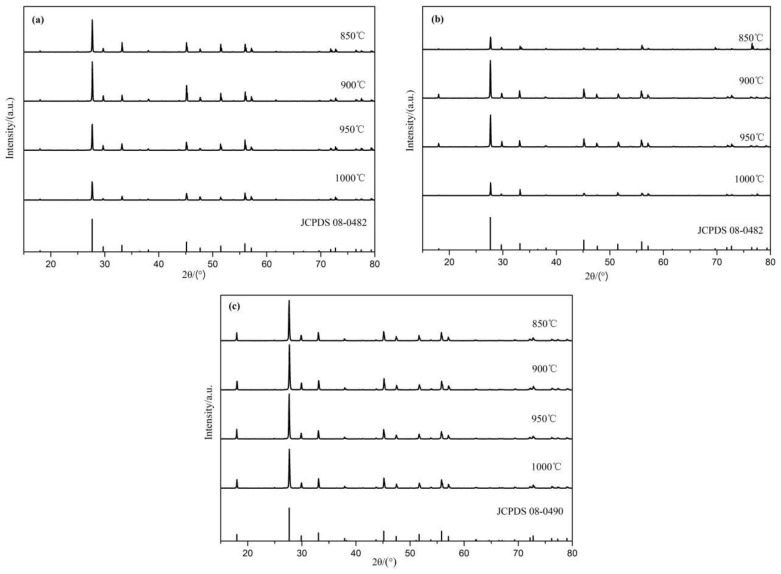
XRD patterns of (**a**) SrMoO_4_, (**b**) SrMo_0.5_W_0.5_O_4_, (**c**) SrWO_4_ synthesized at different temperatures.

**Figure 2 molecules-28-02681-f002:**
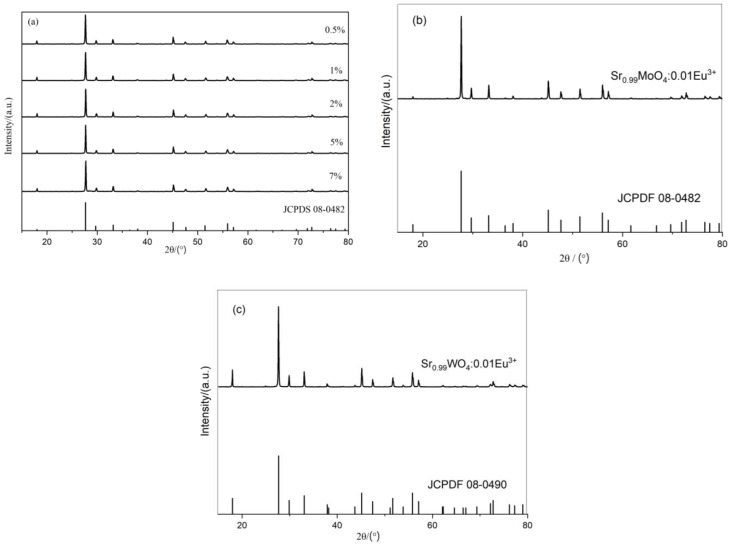
XRD of (**a**) Sr_1−x_Mo_0.5_W_0.5_O_4_:xEu^3+^, (**b**) Sr_0.99_MoO_4_:0.01Eu^3+^ and (**c**) Sr_0.99_WO_4_:0.01Eu^3+^.

**Figure 3 molecules-28-02681-f003:**
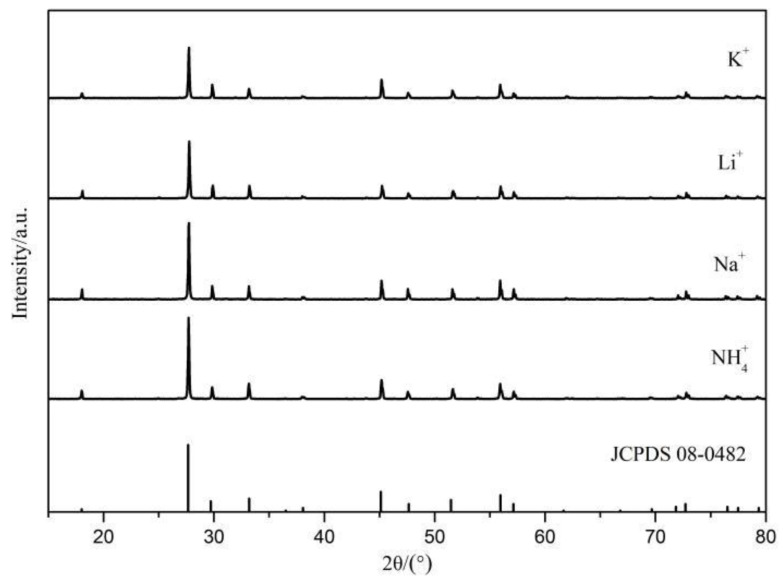
XRD of Sr_0.99_Mo_0.5_W_0.5_O_4_:0.01Eu^3+^ with different charge compensators.

**Figure 4 molecules-28-02681-f004:**
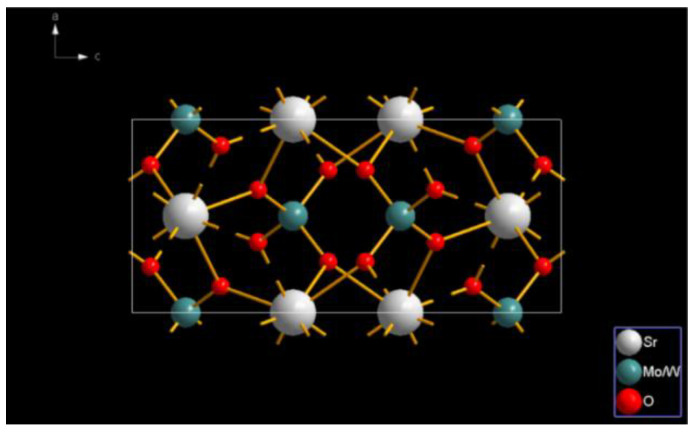
Crystal structure of SrMo_0.5_W_0.5_O_4_.

**Figure 5 molecules-28-02681-f005:**
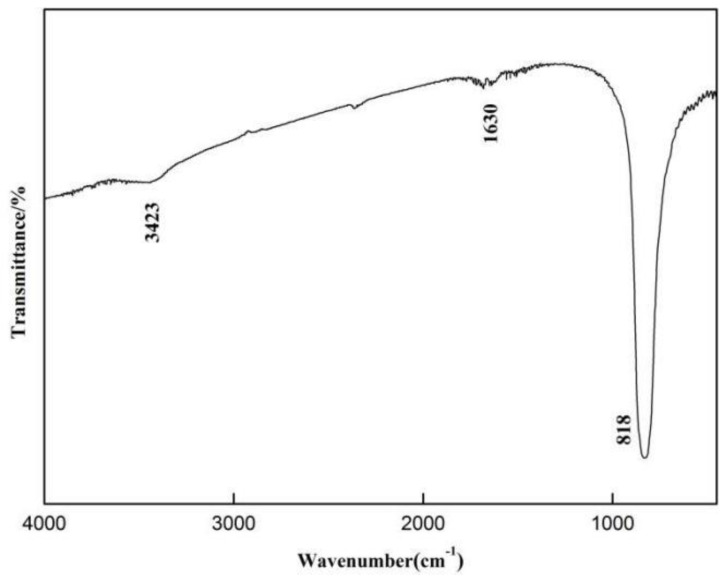
FT-IR spectrum of SrMo_0.5_W_0.5_O_4_ sample.

**Figure 6 molecules-28-02681-f006:**
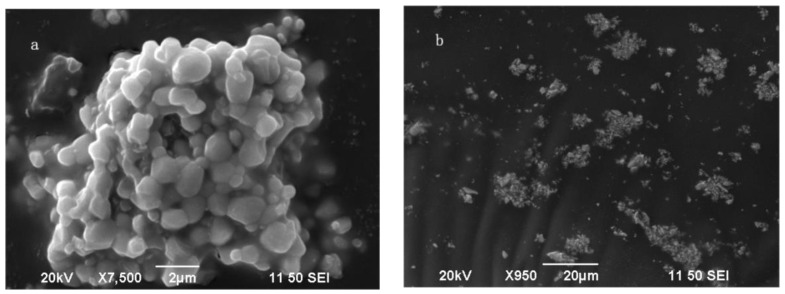
SEM of Sr_0.99_Mo_0.5_W_0.5_O_4_:Eu^3+^. (**a**) 7500 times; (**b**) 950 times.

**Figure 7 molecules-28-02681-f007:**
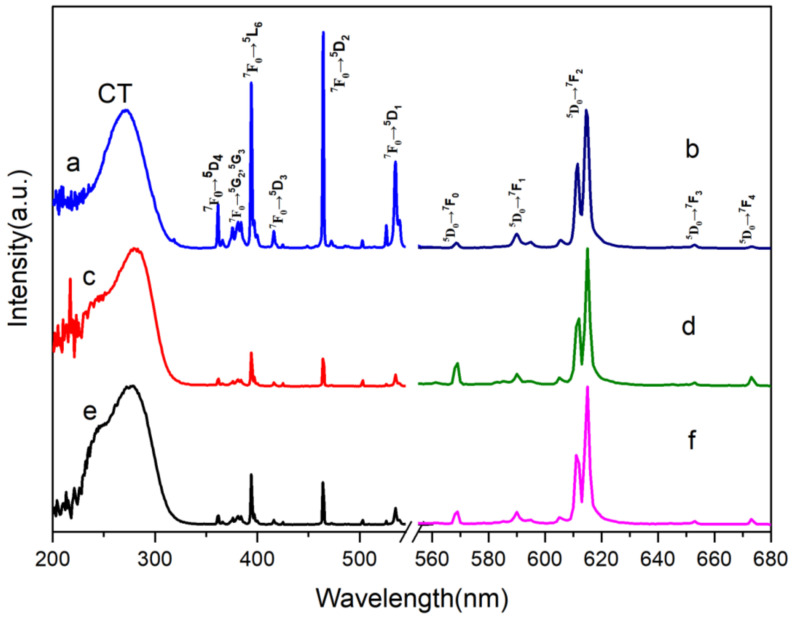
Excitation (**a**,**c**,**e**) and emission (**b**,**d**,**f**) spectra of SrWO_4_:Eu^3+^, SrMo_0.5_W_0.5_O_4_: Eu^3+^, and SrMoO_4_:Eu^3+^.

**Figure 8 molecules-28-02681-f008:**
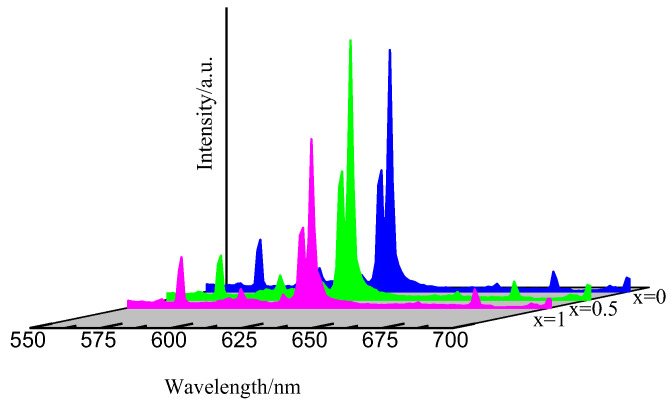
The emission spectrum of Sr_0.99_Mo_x_W_1−x_O_4_:0.01Eu^3+^(x = 0, 0.5, 1).

**Figure 9 molecules-28-02681-f009:**
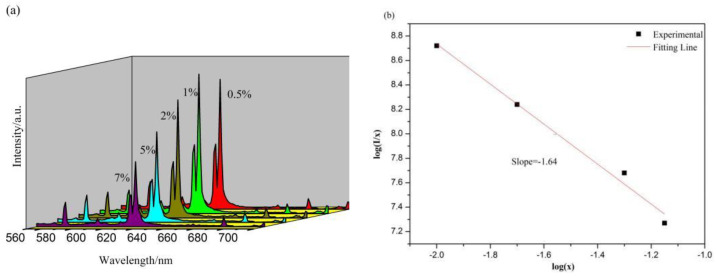
(**a**) The emission spectrum of Sr_1−y_Mo_0.5_W_0.5_O_4_:yEu^3+^(y = 0.005, 0.01, 0.02, 0.05, 0.07), (**b**) Dependence of log (I/x) on log (x) for Sr_1y_Mo_0.5_W_0.5_O_4_:yEu^3+^.

**Figure 10 molecules-28-02681-f010:**
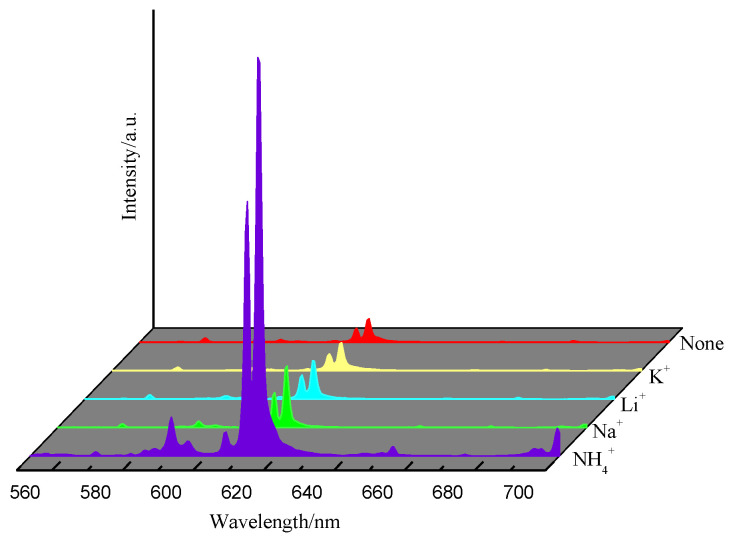
Emission spectra of SrMo_0.5_W_0.5_O_4_:Eu^3+^ with different charge compensators.

**Figure 11 molecules-28-02681-f011:**
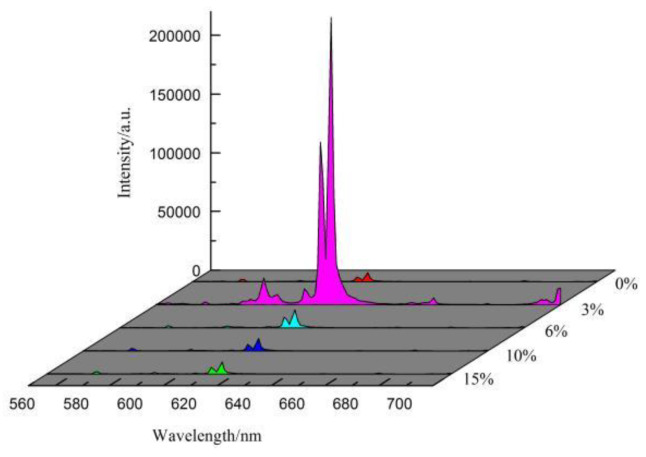
Emission spectra of SrMo_0.5_W_0.5_O_4_:Eu^3+^ with different concentrations of NH_4_^+^.

**Figure 12 molecules-28-02681-f012:**
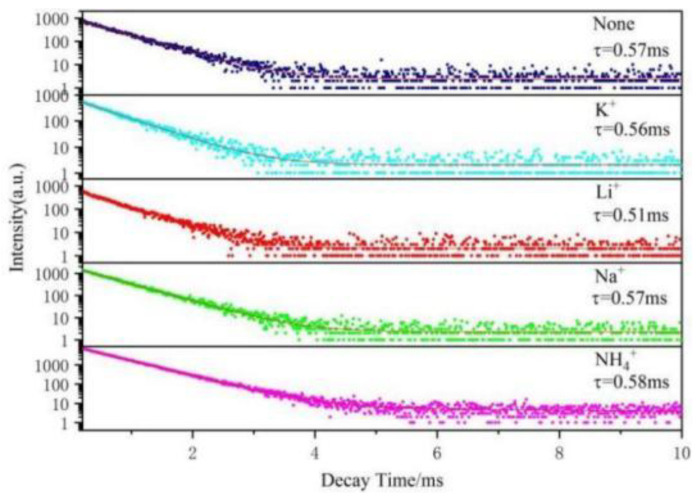
Lifetime decay curve of SrMo_0.5_W_0.5_O_4_:0.01Eu^3+^, A (A = Li^+^, Na^+^, K^+^, NH_4_^+^).

**Figure 13 molecules-28-02681-f013:**
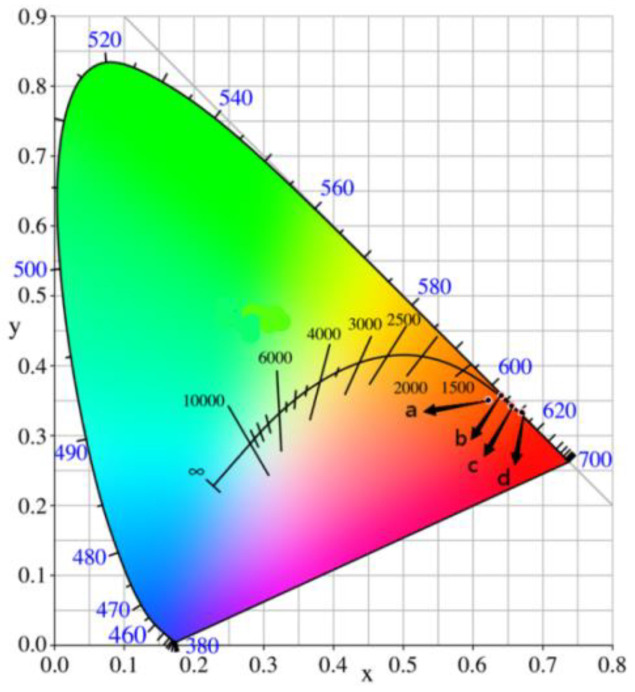
CIE color coordinates of Y_2_O_2_S:Eu^2+^ (**a**), Sr_0.99−x_Mo_0.5_W_0.5_O_4_:0.01Eu^3+^ (**b**), Sr_0.99_Mo_0.5_W_0.5_O_4_:0.01Eu^3+^, 0.03NH_4_^+^ (**c**) and ideal red light (**d**).

## Data Availability

The data presented in this study are available on request from the corresponding author.

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
