# Peer review of "Luminescent Properties and Charge Compensator Effects of SrMo0.5W0.5O4:Eu3+ for White Light LEDs"

_molecules, 2023, doi:10.3390/molecules28062681_

Round 1

Reviewer 1 Report

The work has a clearly expressed practical orientation. The study showed that the red phosphors synthesized by the high-temperature solid-phase technique of SrMo0.5W0.5O4:Eu3+ using NH4+ charge compensators have promising applications as a red phosphor that can be successfully stimulated by blue light and ultraviolet light.

 However, the FT-IR spectrum of the SrMo0.5W0.5O4 sample (figure 5) should preferably be presented along the x-axis in nanometers, as well as on all the excitation and emission spectra (figures 7-11).

Author Response

Review report(1):

The work has a clearly expressed practical orientation. The study showed that the red phosphors synthesized by the high-temperature solid-phase technique of SrMo0.5W0.5O4:Eu3+ using NH4+ charge compensators have promising applications as a red phosphor that can be successfully stimulated by blue light and ultraviolet light.

 (1)However, the FT-IR spectrum of the SrMo0.5W0.5O4 sample (figure 5) should preferably be presented along the x-axis in nanometers, as well as on all the excitation and emission spectra (figures 7-11).

â–²In general, the unit of The x-axis of FT-IR spectrum uses the wavenumber, while the x-axis for excitation and emission spectra use the wavelength, so the x-axis of Figure 5 in the text is not the same as in Figure 7-11.

Reviewer 2 Report

The article by Kong et al is devoted to the creating luminescent materials based on Eu3+.
Unfortunately, the article cannot be published in its present form. In my opinion, it should be rewritten almost completely and submitted anew after a thorough revision.

Abstract, line 14:
suitable red fluorescent material for UV LED and blue light LED

Bad wording. Apparently it was implied that “fluorescent material suitable for excitement with UV LED or Blue LED”?

Line 15-16 “The ideal Eu3+ doping concentration in SrMo0.5W0.5O4:Eu3+ phosphor is 1%, with Rc = 25.57 Å as the critical distance

What was meant?

Line 19, line 20: NH4+ - upper and lower indexes should be checked throughout the text

Introduction - the section is poorly written. There are few references, many statements are not accompanied by references. English requires correction. The second paragraph of the Introduction is also written very strangely. Several phosphors are listed with a short description of their properties, namely the compounds Sr3NaSbO6:Eu3+, Li2.06Nb0.18Ti0.76O3:Eu3+, NaY1-xEuxGeO4, NaBaBi2(PO4)3:Eu3+ and Cs3EuGe3O9. The choice of such examples is very strange. Each of the examples is rare (especially germanium compounds), and does not represent a representative picture of Eu2O3-bades phosphors. In addition to these compounds, there are many hundreds of onother studied Eu-luminophors. In this case, it should be indicated how the compounds listed in the Introduction are better than others

The experimental part:

Which oven was used? In what atmosphere was the synthesis carried out (air, O2, N2)? How precisely was the temperature controlled?

In what mode were the luminescence decay curves recorded and deconvoluted?

Results and discussion:

Has the PXRD curves been indexed? If not, then one need to perform this operation and bring a table with the parameters of the unit cell. The PXRD curves have an auxiliary function and most of these pictures can be transferred to a Supporting Information file.

The IR spectrum was registered with poor-quality subtraction of the baseline. Does the position and shape of the Mo=O (W=O) oscillation lines change at 818 cm-1? This question can be answered by the IR spectra of other samples not given in the work.

«Figure 6 shows that the phosphor Sr0.99Mo0.5W0.5O4:0.01Eu3+ has good crystallinity»
From the reviewer's point of view, Figure 6 indicates POOR crystallinity

Figure 7, luminescence spectra:
1) All transitions in the EXCITATION and EMISSION spectra must be attributed!
2) In the above form, it is extremely difficult to compare the spectra. Arrange the same type of images on top of each other on the same spectrum, obtaining two figures: the excitation spectra of samples a, b, c, etc., as well as the emission spectra of samples a, b, c, etc.

Line 215, line 227 etc: "The intensity of the emission increases after the addition of Mo6+ ions 215 to SrWO4:Eu3+" Mo6+ ions are impossible in chemistry, except in a charged particle accelerator or in the solar wind. We can talk about the substitution of tungsten (VI) with molybdenum (VI), but not about such ions!

How was the luminescence intensity measured? Has the absolute quantum yield been measured? What equipment and optical scheme were used?

Figures 8-10 are very difficult to perceive. Do you need to compare the intensity? This is not possible in the current view. How does the form of splitting of Europium transitions correlate with the Stark effect? It is impossible to see.
How can ammonium ions compensate for the charge if ammonium compounds are not thermally stable?
The luminescence attenuation curves in all cases, except possibly the first one (“none”), are not explicitly described by a monoexponent. Try a bi-exponential description!

Author Response

Review report(2):

The article by Kong et al is devoted to the creating luminescent materials based on Eu3+. Unfortunately, the article cannot be published in its present form. In my opinion, it should be rewritten almost completely and submitted anew after a thorough revision.

(1)Abstract, line 14:“suitable red fluorescent material for UV LED and blue light LED” Bad wording. Apparently it was implied that “fluorescent material suitable for excitement with UV LED or Blue LED”?

â–² This sentence has been revised in the manuscript.

(2)Line 15-16 “The ideal Eu3+ doping concentration in SrMo0.5W0.5O4:Eu3+ phosphor is 1%, with Rc = 25.57 Å as the critical distance”, What was meant?

â–² It is revised in the manuscript.

(3)Line 19, line 20: NH4+ - upper and lower indexes should be checked throughout the text

 â–² The upper and lower indexes of NH4have been revised in the manuscript, and have been corrected carefully throughout the text.

.(4)Introduction - the section is poorly written. There are few references, many statements are not accompanied by references. English requires correction. The second paragraph of the Introduction is also written very strangely. Several phosphors are listed with a short description of their properties, namely the compounds Sr3NaSbO6:Eu3+, Li2.06Nb0.18Ti0.76O3:Eu3+, NaY1-xEuxGeO4, NaBaBi2(PO4)3:Eu3+ and Cs3EuGe3O9. The choice of such examples is very strange. Each of the examples is rare (especially germanium compounds), and does not represent a representative picture of Eu2O3-bades phosphors. In addition to these compounds, there are many hundreds of onother studied Eu-luminophors. In this case, it should be indicated how the compounds listed in the Introduction are better than others

â–²The first paragraph of the introduction is about LED phosphor, the second paragraph is about Eu3+ red phosphors, the third paragraph is about Eu3+ -doped molybdenum tungstate phosphor, the last paragraph points out the content of this article. Five references were added to the introduction, and two references were replaced in the revised manuscript.

(5)The experimental part:

Which oven was used? In what atmosphere was the synthesis carried out (air, O2, N2)? How precisely was the temperature controlled?

â–²Synthesized phosphors used high-temperature furnace in air atmosphere have been seen in the manuscript,

(6)In what mode were the luminescence decay curves recorded and deconvoluted?

â–²The measuring and test instruments recorded the luminescence decay curves has been provided in the revised manuscript.

(7)Results and discussion:

Has the PXRD curves been indexed? If not, then one need to perform this operation and bring a table with the parameters of the unit cell. The PXRD curves have an auxiliary function and most of these pictures can be transferred to a Supporting Information file.

â–²The synthesized SrMo0.5W0.5O4 and SrMoO4 (PDF 08-0482) parameters are shown in the table below.

Parameters 

Synthesized

PDF08-0482

a

5.3907

5.3944

b

5.3907

5.3944

c

12.0118

12.0200

α

90

90

β

90

90

γ

90

90

denaity

4.7098

4.7000

vol

349.10

349.78

(8)The IR spectrum was registered with poor-quality subtraction of the baseline. Does the position and shape of the Mo=O (W=O) oscillation lines change at 818 cm-1? This question can be answered by the IR spectra of other samples not given in the work.

â–² The infrared spectra of SrMoO4, SrWO4, SrMo0.5W0.5O4 is basically the same, as shown in the figure below.

(9)«Figure 6 shows that the phosphor Sr0.99Mo0.5W0.5O4:0.01Eu3+ has good crystallinity»

From the reviewer's point of view, Figure 6 indicates POOR crystallinity

â–²It has been revised in the text.

(10)Figure 7, luminescence spectra:

1) All transitions in the EXCITATION and EMISSION spectra must be attributed!

2) In the above form, it is extremely difficult to compare the spectra. Arrange the same type of images on top of each other on the same spectrum, obtaining two figures: the excitation spectra of samples a, b, c, etc., as well as the emission spectra of samples a, b, c, etc.

â–²All transitions in the excitation and emission spectra have been attributed, the figure7 has been redrawn in the text.

(11)Line 215, line 227 etc: "The intensity of the emission increases after the addition of Mo6+ ions 215 to SrWO4:Eu3+Mo6+ ions are impossible in chemistry, except in a charged particle accelerator or in the solar wind. We can talk about the substitution of tungsten (VI) with molybdenum (VI), but not about such ions!

â–²W (â…¥) and Mo (â…¥)  have been replaced to W6+ and Mo6+ in the revised manuscript.

(12)How was the luminescence intensity measured? Has the absolute quantum yield been measured? What equipment and optical scheme were used?

â–²The luminescence intensities were evaluated using an Edinburgh FS5 fluorescence spectrometer.  It's so regret that the absolute quantum yield can't be measured in our lab.

(13)Figures 8-10 are very difficult to perceive. Do you need to compare the intensity? This is not possible in the current view. How does the form of splitting of Europium transitions correlate with the Stark effect? It is impossible to see.

â–²The charge compensators improves the luminescence intensity, and their luminescence center ion transition does not affect.

  • How can ammonium ions compensate for the charge if ammonium compounds are not thermally stable?

â–²Ammonium root content is less and relatively stable in the phosphors.

(15)The luminescence attenuation curves in all cases, except possibly the first one (“none”), are not explicitly described by a monoexponent. Try a bi-exponential description!

â–²The luminescence decay of Eu3+ is a bi-exponential description! The values in the figure are the average lifetime obtained from the double-exponential description. It has been revised in the text.

â—† In addition, we have made some other changes besides above. For details, see the new version.
